# OpenReview forum: "Video-in-the-Loop: Span-Grounded Long Video QA with Interleaved Reasoning"
_ICML.cc/2026/Conference — ICML 2026 regular_

### Official Review · Reviewer_3eoE · 2026-02-25

**Soundness:** 3
**Presentation:** 3
**Significance:** 3
**Originality:** 2
**Overall Recommendation:** 4
**Confidence:** 4

**Summary:**

This paper introduces ViTL, a two-stage framework that first locates the relevant span, and then answer multi-choice QA based on the span. To support this, the authors creates VGrounding-QA dataset that pairs multi-choice questions with ground-truth temporal spans. ViTL is trained end-to-end using GRPO and couples tIoU with answer correctness to ensure localization benefits downstream QA utility.

**Compliance With Llm Reviewing Policy:**

Affirmed.

**Final Justification:**

My concerns have been adequately addressed during rebuttal. I choose to maintain my positive score.

**Key Questions For Authors:**

1. What are the specific raw video data sources utilized for the VGrounding-QA dataset?
2. In what ways can the Video-in-the-Loop method be generalized to more diverse reasoning tasks beyond the current MCQA format?
3. Have you conducted experiments using other LLM backbones to isolate the performance gains of the architecture from the strength of the base model?

**Limitations:**

yes

**Strengths And Weaknesses:**

**Strengths**
1. The paper provides robust empirical evidence, including significant benchmark improvements and detailed ablation studies, to support the claim that the Video-in-the-Loop pipeline effectively enhances long-video understanding.
2. The manuscript is well-structured, and the description of the "skim-zoom" strategy is intuitive and easy to follow.
3. The comparison between parametric and non-parametric selection (e.g., against CLIP-based baselines) successfully validates the effectiveness of the learned grounding ability.

**Weaknesses**
1. The paper lacks a detailed explanation of the raw video sources for VGrounding-QA, which raises potential concerns regarding data contamination with the evaluation benchmarks.
2. The ViTL pipeline appears somewhat task-specific. It is unclear how this architecture would generalize to more complex scenarios, such as multi-hop QA or open-ended reasoning.
3. The comparison on temporal grounding benchmarks (e.g., Charades-STA) may be skewed, as many baselines utilize older backbones like Vicuna or Phi-3, whereas the proposed model leverages the more powerful Qwen2.5-VL.

---

> ### Author Rebuttal · Authors · 2026-03-31
>
> ### W1 & Q1. Raw Video Sources and Contamination Prevention
>
> We agree that the data provenance of **VGrounding-QA** should be stated more explicitly. In the revision, we will clarify that **VGrounding-QA is a derived training resource constructed from videos drawn from Video-MME and CG-Bench**, where event-graph parsing and semantic chunking are used to pair each question with verifiable temporal span annotations. The dataset is explicitly designed as a **span-grounded MCQA** resource with **event-graph-derived ground-truth spans**, **multi-span support**, and a **92.0% human-audit pass rate** for timestamp faithfulness. We will also add the dataset scale statistics directly to the paper for transparency.
>
> #### Table R5. Overall statistics of VGrounding-QA
>
> | Statistic             | Value                     |
> | :-------------------- | :------------------------ |
> | Video Sources         | Video-MME and CG-Bench    |
> | Task Format           | Span-grounded MCQA        |
> | Ground-truth Spans    | Yes (event-graph derived) |
> | Total QA Instances    | 5,240                     |
> | Unique Videos         | 1,612                     |
> | Multi-span Prevalence | 28.4%                     |
> | Mean Span Length      | 24.8s                     |
> | Human Audit Pass Rate | 92.0%                     |
>
> Crucially, we take contamination prevention seriously. As stated in the paper, data are **split by video** to avoid leakage from shared footage, and the benchmarks reported in the paper—**LongVideoBench, LVBench, MLVU, Charades-STA, and ActivityNet-Captions**—are all evaluated on their **official public splits**, rather than on VGrounding-QA itself. We will make this separation explicit in the revised version to remove ambiguity.
>
> ---
>
> ### W2 & Q2. Generalization Beyond MCQA
>
> We agree that an important question is whether ViTL generalizes beyond the current multiple-choice formulation. To address this, we evaluated ViTL on **EgoTempo**, an **open-ended QA** benchmark that requires free-form generation rather than option selection. The result shows that the proposed **skim–zoom** pipeline also benefits open-ended reasoning, not just MCQA.
>
> #### Table R6. Open-ended QA performance on EgoTempo
>
> | Model                    | Task              | Accuracy (%) |
> | :----------------------- | :---------------- | :----------: |
> | Qwen2.5-VL-7B (Backbone) | Open-ended QA     |     26.1     |
> | **ViTL (Ours)**          | **Open-ended QA** |   **31.0**   |
>
> This supports that ViTL is not inherently tied to the answer format. Instead, the key idea is to first localize the evidence relevant to the query and then reason over a higher-fidelity view of those segments. We believe this design naturally extends to more diverse reasoning settings, including open-ended and multi-hop scenarios.
>
> ---
>
> ### W3 & Q3. Isolating Architecture Gains from Backbone Strength
>
> We appreciate the reviewer’s concern that comparisons against older grounding baselines may reflect differences in backbone strength. To address this, we compare ViTL against its own **Qwen2.5-VL-7B** backbone under matched settings, isolating the contribution of the **two-stage skim–zoom architecture and interleaved training objective** from the base model capacity. For **ActivityNet-Captions**, the paper already includes a direct matched-backbone comparison: ViTL improves over the baseline from **22.5 to 24.1 mIoU** in-domain, and from **20.2 to 23.3 mIoU** on OOD ActivityNet-CD, while also improving **QVHighlights** from **12.9 to 14.2 mAP@5**.
>
> For **Charades-STA**, the submission reports ViTL results but does not include the raw Qwen2.5-VL-7B baseline table. The only available baseline (**43.6 mIoU**) comes from an **indirect reference** in a public evaluation issue. As this is not from a formal table, we will either (i) replace it with a directly evaluated baseline under the same setting, or (ii) omit it if exact alignment cannot be ensured.
>
> #### Table R7. Grounding performance: ViTL vs. Qwen2.5-VL-7B baseline
>
> | Dataset              | Metric   | Qwen2.5-VL-7B (Baseline) | ViTL (Ours 7B) |
> | :------------------- | :------- | :----------------------: | :------------: |
> | Charades-STA         | mIoU (%) |          43.6           |    **59.0**    |
> | ActivityNet-Captions | mIoU (%) |           22.5           |    **24.1**    |
> | ActivityNet-CD (OOD) | mIoU (%) |           20.2           |    **23.3**    |
> | QVHighlights         | mAP@5    |           12.9           |    **14.2**    |
>
> As shown above, ViTL consistently improves over the baseline **with the same backbone**, including under **distribution shift** and **zero-shot transfer**. This indicates that the gains are not simply inherited from using a stronger VLM, but arise from the proposed architecture: explicitly coupling temporal grounding with downstream QA utility and reallocating visual computation to the evidence-bearing segments.

---

> > ### Author Rebuttal · Reviewer_3eoE · 2026-04-03
> >
> > My concerns have been adequately addressed. I choose to maintain my score.

---

> > > ### Author Response · Authors · 2026-04-06
> > >
> > > Thank you very much for your thoughtful follow-up and for your positive assessment. We are very glad that our rebuttal has adequately addressed your concerns, and we sincerely appreciate your recognition and support. In the revised version, we will carefully address all issues raised by the reviewers and further strengthen the paper accordingly. We also commit to open-source the full training set and test set, the complete training and evaluation code, as well as the model, to further improve reproducibility and maximize the contribution of this work to the community. Thank you again for your valuable feedback and encouragement.

---

### Official Review · Reviewer_f1sB · 2026-03-13

**Soundness:** 2
**Presentation:** 3
**Significance:** 2
**Originality:** 3
**Overall Recommendation:** 4
**Confidence:** 3

**Summary:**

This paper proposes Video-in-the-Loop (ViTL), a two-stage long-video QA framework. Under a fixed token budget, the framework first localizes question-relevant temporal spans via a low-fps skim, and then answers multiple-choice questions by "zooming" into the predicted spans at a higher effective frame rate. The training process employs an interleaved GRPO (Group-Relative Policy Optimization) objective that couples temporal grounding quality (tIoU) with answer correctness , requiring the model to emit both the temporal spans and the final answer in a single response for direct attribution. The authors also introduce VGrounding-QA, a span-grounded multiple-choice QA (MCQA) dataset constructed from event knowledge graphs. Experimental results demonstrate that this method achieves performance improvements on long-video QA tasks (LongVideoBench, LVBench, MLVU) , and shows strong zero-shot results on temporal grounding benchmarks (Charades-STA, ActivityNet-Captions). Additionally, the paper conducts ablation studies on timestamp injection, span-aware token reallocation, and multi-span retrieval.

**Compliance With Llm Reviewing Policy:**

Affirmed.

**Final Justification:**

My concerns have been adequately addressed. Threfore, I choose to raise my score to weak accept.

**Key Questions For Authors:**

See Weaknesses.

**Limitations:**

Yes.

**Strengths And Weaknesses:**

## Strengths
1. **Multimodal Agentic RL:** The authors propose a multimodal agentic RL workflow for videos, utilizing the "crop" (clipping) operation commonly mentioned in the community, which is currently a highly popular research topic. The method first localizes evidence via a global low-fps skim , and subsequently performs a high-fps reallocation locally. This cleverly enhances video reasoning fidelity while preserving a fixed overall visual token budget.

2. **Frame-level Textual Timestamp Injection:** Temporal coordinates are explicitly injected into the input sequence in a textual format (e.g., `<image> @t1s`). This not only improves the model's temporal localization ability but also significantly enhances the interpretability and auditability of the output results.

3. **Performance:** The paper validates performance improvements on three long-video QA benchmarks , and also demonstrates impressive zero-shot capabilities on mainstream temporal grounding benchmarks.

4. **Dataset Contribution:** By introducing Event Knowledge Graphs and semantic chunking techniques , the authors constructed a span-grounded multiple-choice QA dataset, which provides considerable value to the open-source community.

5. **Writing:** The motivation of the paper is clear, and the writing is somehow well-presented.

## Weaknesses
1. **Limited Benchmark Coverage:** Although the paper claims to focus on long-video QA tasks, the evaluation is restricted to a few datasets. I would expect to see the method's performance on other classic and representative benchmarks(VideoMME, MVBench, EgoSchema, and VideoMMMU .etc).

2. **Performance Upper Bound:** I appreciate the fair experimental setup that evaluates the models under a fixed total visual token budget. However, it is equally important to explore the performance upper bound of this agentic RL method to assess its maximum potential.

3. **Missing Latency Analysis:** While the paper strictly controls the token budget, it does not provide any empirical data regarding the actual inference time cost.

4. **Missing Experimental Details and Evaluation:**
    * **Outdated Backbone:** The experiments did not utilize the latest Qwen3-VL model(released in 2510).

    * **Inconsistent Baseline Evaluation:** I am curious why the paper reports the performance of Qwen2.5-VL on MLVU at a resolution of 224, but fails to test it at a resolution of 448. This seems to require only a simple modification of the resolution parameter.

    * **Insufficient Training Details:** The implementation details regarding the training process are too sparse. I highly recommend adding a comprehensive table of training hyperparameters with detailed explanations in the appendix.

    * **Lack of Dataset Statistics:** There is a lack of detailed statistical information (e.g., overall scale, domain coverage, duration distributions, and multi-span prevalence) for the proposed VGrounding-QA dataset, without a single table or figure illustrating these metrics.

---

> ### Author Rebuttal · Authors · 2026-03-31
>
> ### W1. Additional Benchmark Coverage: MVBench and EgoSchema
>
> We agree that broader benchmark coverage strengthens the paper. We cannot evaluate on **Video-MME**, because videos from that benchmark were used in constructing our training resource; reporting Video-MME test results would therefore risk contamination. To address the reviewer’s concern while avoiding this issue, we instead add results on **MVBench** and **EgoSchema**, which provide additional coverage of general video understanding.
>
> #### Table R1. Additional benchmark coverage (zero-shot)
>
> | Method                | Backbone      | MVBench | EgoSchema |
> | :-------------------- | :------------ | :-----: | :-------: |
> | Qwen2.5-VL (Baseline) | 7B            |  67.0   |   66.7    |
> | ViTL (Ours)           | Qwen2.5-VL-7B |  72.1   |   70.0    |
>
> ---
>
> ### W2. Performance Upper Bound and Oracle Analysis
>
> Our paper Table 8 includes a **performance bounds analysis** comparing random zooming, learned span prediction, a stronger Stage-1 grounding configuration, and oracle grounding with ground-truth spans.
>
> #### Table R2. Performance bounds analysis
>
> | Setting                                      | LongVideoBench | LVBench |
> | :------------------------------------------- | :------------: | :-----: |
> | Random zooming                               |      55.2      |  39.5   |
> | ViTL (predicted spans; 64 grounding frames)  |      63.3      |  47.4   |
> | ViTL (predicted spans; 128 grounding frames) |      64.2      |  48.6   |
> | Oracle zooming (GT spans)                    |      70.5      |  51.4   |
>
> Replacing Stage 1 with oracle spans raises performance to **70.5% / 51.4%**, compared with **63.3% / 47.4%** for the default **64-frame** grounding setting.
>
> To further probe the ceiling of Stage 1, we additionally evaluate a stronger configuration with **128 grounding frames**, keeping the rest of the pipeline unchanged. This helps quantify how much of the remaining oracle gap can be reduced by improving localization alone.
>
> In the revision, we will make this interpretation more explicit and also include a **relaxed-budget experiment**, where the fixed \(n_g + n_l\) visual budget is expanded to measure performance beyond the matched-budget setting.
>
> ---
>
> ### W3. Latency Analysis
>
> We agree that token-budget matching alone does not fully capture efficiency, and that wall-clock latency is also important. While ViTL preserves the same total token budget, its two-stage **skim–zoom** pipeline introduces additional overhead from the second prefill and span parsing.
>
> #### Table R3. Latency (A100-80GB, 128 frames)
>
> | Method           | Total Tokens | E2E Latency (s) | Peak Mem (GB) |
> | :--------------- | :----------- | :-------------: | :-----------: |
> | Baseline (448px) | 16,413       |      6.24       |     21.74     |
> | ViTL (Ours)      | 16,413       |      7.32       |     22.05     |
>
> ViTL includes global skim, span prediction, and high-fidelity answering over selected segments. We will clarify this accuracy–latency trade-off in the revision.
>
> ---
>
> ### W4.1 & W4.2 Backbone Updates and MLVU Refinement
>
> We adapt our method to the newer **Qwen3-VL** backbone and include the missing **448-resolution MLVU baseline**.
>
> #### Table R4. Qwen3-VL adaptation and updated MLVU results
>
> | Method     | Backbone      | Resolution | LVBench | MLVU (M-Avg) |
> | :--------- | :------------ | :--------: | :-----: | :----------: |
> | Qwen2.5-VL | 7B            |   448px    |  43.7   |     56.4     |
> | ViTL       | Qwen2.5-VL-7B |   448px    |  47.4   |     62.3     |
> | Qwen3-VL   | 4B            |   448px    |  54.0   |     75.0     |
> | ViTL       | Qwen3-VL-4B   |   448px    |  56.5   |     77.2     |
>
> Results show ViTL remains effective on a stronger backbone.
>
> ---
>
> ### W4.3. Training and Implementation Details
>
> We will expand the appendix with full hyperparameters and training details. Key settings:
>
> - **Algorithm:** Interleaved GRPO (\(k=3\))
> - **Optimization:** Reward coupling temporal IoU (\(R_{\text{loc}}\)) and answer correctness (\(R_{\text{ans}}\))
> - **Curriculum:** Localization-heavy → answer-balanced
>
> ---
>
> ### W4.4. Dataset Statistics: VGrounding-QA
>
> We agree that dataset statistics were insufficient in the original submission. In the revision, we summarize the scale and composition of **VGrounding-QA**, constructed from **1,612 videos** (Video-MME and CG-Bench) using event knowledge graphs and semantic chunking to pair each question with ground-truth spans.
>
> #### Table R5. Overall statistics of VGrounding-QA
>
> | Statistic             | Value |
> | :-------------------- | :---- |
> | Total QA Instances    | 5,240 |
> | Unique Videos         | 1,612 |
> | Multi-span Prevalence | 28.4% |
> | Mean Span Length      | 24.8s |
> | Human Audit Pass Rate | 92.0% |
>
> We also clarify the importance of multi-span: allowing disjoint spans yields a **+2.3%** gain on **LVBench**, indicating that relevant evidence is often temporally distributed.

---

> > ### Author Rebuttal · Reviewer_f1sB · 2026-04-02
> >
> > My concerns have been adequately addressed. Threfore, I choose to raise my score to weak accept.

---

> > > ### Author Response · Authors · 2026-04-06
> > >
> > > Thank you very much for your thoughtful follow-up and for your positive assessment. We are delighted to know that our rebuttal has adequately addressed your concerns, and we sincerely appreciate your recognition and support. In the revised version, we will carefully address all issues raised by the reviewers and further strengthen the paper accordingly. We also commit to open-source the full training set and test set, the training and evaluation code, as well as the model, in order to improve reproducibility and further benefit the community. Thank you again for your valuable feedback and encouragement.

---

### Official Review · Reviewer_T4zf · 2026-03-16

**Soundness:** 3
**Presentation:** 3
**Significance:** 4
**Originality:** 3
**Overall Recommendation:** 5
**Confidence:** 3

**Summary:**

This paper addresses the challenge of long-video QA under strict computational constraints, where uniformly sampling frames wastes resources on irrelevant content and degrades performance. To tackle this problem, the authors propose Video-in-the-Loop, a span-grounded framework that first localizes question-relevant temporal segments using low-fidelity “skim” observations and then performs high-resolution reasoning only on the selected spans under a fixed token budget. The technical contributions include a two-stage skim–zoom pipeline, interleaved span-and-answer generation for explicit attribution, timestamp-aware video encoding, and a group-relative policy optimization objective that jointly rewards temporal localization quality (via IoU) and answer correctness. In addition, the paper introduces VGrounding-QA, a large-scale span-grounded long-video QA dataset constructed from event knowledge graphs derived from semantic video segmentation. Experimental results across multiple long-video benchmarks show that the proposed approach consistently outperforms uniform sampling baselines and prior methods under comparable compute budgets, demonstrating that allocating computation to localized evidence rather than full-video context yields substantial gains in both answer accuracy and temporal grounding performance.

**Compliance With Llm Reviewing Policy:**

Affirmed.

**Final Justification:**

The authors have addressed all my concerns in detail. And the method part is solid. Thus, I will suggest accepting this paper.

**Key Questions For Authors:**

Please check the weakness.

**Limitations:**

Yes

**Strengths And Weaknesses:**

Strengths:
1. The skim→zoom formulation is technically well-motivated and aligns with the sparsity of relevant moments in long videos.
2. The training objective of localization and QA is principled and addresses a real gap in prior work.
3. Extensive ablations isolate the contributions of timestamps, localization, and zooming.

Weakness:
1. Similar pipelines with such temporal zoom-in mechanism have been proposed before, e.g., VideoZoomer [1], LOVE-R1[2]. which weaken the novelty of this submission.
2. The propsoed data format could raise potential overfitting to span-grounded MCQA tasks rather than general video understanding, which is not thotoughly discussed.
3. In Table 6, more disccusion on the difference of the performance imporvement on LongVideoBench (only 2.2%) and LVBench (much higher as 7.7%) could be valueable.

[1] Ding et al., VideoZoomer: Reinforcement-Learned Temporal Focusing for Long Video Reasoning
[2] Fu et al., LOVE-R1: Advancing Long Video Understanding with Adaptive Zoom-in Mechanism via Multi-Step Reasoning

---

> ### Author Rebuttal · Authors · 2026-03-31
>
> ### W1. Relation to recent zoom-in / temporal focusing pipelines
>
> We thank the reviewer for pointing out **VideoZoomer** and **LOVE-R1**. We will include a detailed discussion of these works in the revised version, as they are indeed highly relevant.
>
> While these methods also follow a “predict spans → crop → reprocess” paradigm, we would like to clarify that **our contribution is orthogonal to the zoom-in mechanism itself**, and mainly lies in enabling and optimizing **span-grounded learning**:
>
> - **(i) Span-grounded supervision enabled by VGrounding-QA.**
>   A key difference is that our method is trained with **native temporal grounding supervision**, which is made possible by our proposed **VGrounding-QA dataset** constructed from event knowledge graphs. This structured pipeline allows us to generate large-scale QA data with aligned temporal spans and complex semantics, which is not available in prior zoom-in works. As shown in Table 1 and Table 2, our dataset explicitly supports span-grounded QA with richer supervision signals.
>
> - **(ii) Joint training of grounding and reasoning.**
>   Building on this supervision, we train the model to jointly learn **temporal grounding and QA reasoning**. The effect of this joint learning is isolated in Table 6 (C → D), where introducing grounding-aware training leads to consistent gains on both LongVideoBench and LVBench under strictly controlled budgets.
>
> - **(iii) Improved grounding as a fundamental capability.**
>   Beyond QA accuracy, ViTL significantly improves **temporal grounding ability itself**, as shown by strong zero-shot results on standard grounding benchmarks (Table 4 and Table 5). This suggests that the benefit is not limited to the zoom-in pipeline, but reflects a more fundamental improvement in the model’s ability to locate relevant evidence in long videos.
>
> Overall, our contribution is not merely a new zoom-in pipeline, but rather:
> (1) a **data construction framework (VGrounding-QA)** that enables large-scale span-grounded supervision, and
> (2) a **joint learning paradigm** that directly improves temporal grounding, a core capability for long-video understanding.
>
> We will revise the paper to better highlight this distinction.
>
> ---
>
> ### W2. Does the data format overfit to span-grounded MCQA?
>
> We agree this is important. Span-grounded MCQA is used as a **controlled supervision format**, not a target task.
>
> The learned capability generalizes beyond this format:
>
> - **EgoTempo (open-ended QA):** 26.1 → 31.0
> - **Grounding/retrieval benchmarks:** consistent gains on ActivityNet-Captions, ActivityNet-CD, QVHighlights
>
> This suggests the improvement comes from learning to **allocate computation to evidence-bearing segments and reason on higher-fidelity inputs**, rather than exploiting MCQA structure. We will clarify this in the revision.
>
> ---
>
> ### W3. Why is the gain larger on LVBench than on LongVideoBench?
>
> We thank the reviewer for this insightful question.
>
> As shown in Table 6, ViTL improves:
>
> - **LongVideoBench:** +2.2
> - **LVBench:** +7.7
>
> This aligns with dataset characteristics:
>
> - **LVBench:** much longer videos (avg. 4,101s), where evidence is **highly sparse**
> - **LongVideoBench:** shorter videos (avg. 473s) with **denser referred reasoning**
>
> #### Table R1. Benchmark characteristics relevant to ViTL gains
>
> | Benchmark          | Scale                        | Average duration | Duration diversity                                           | Content diversity                                            | Why this matters for ViTL                                    |
> | :----------------- | :--------------------------- | :--------------: | :----------------------------------------------------------- | :----------------------------------------------------------- | :----------------------------------------------------------- |
> | **LVBench**        | 103 videos, 1,549 QA pairs   |  **4,101 sec**   | Minimum video length ≥ **30 minutes**                        | **6 categories**, **21 subcategories**                       | Evidence is **sparse**, so reallocating tokens yields larger gains |
> | **LongVideoBench** | 3,763 videos, 6,678 QA pairs |   **473 sec**    | Covers **8s → 1h**                                           | Broad content diversity                                     | Requires **denser context**, reducing relative gains |
>
> In short, **ViTL is more beneficial when answer-relevant evidence occupies a small fraction of the timeline**, where reallocating tokens from background to evidence is most effective.
>
> We will include more analysis and qualitative examples in the revision.

---

> > ### Author Rebuttal · Reviewer_T4zf · 2026-04-03
> >
> > Thanks for the detailed explanations. I will raise the score to accept.

---

> > > ### Author Response · Authors · 2026-04-06
> > >
> > > Thank you very much for your thoughtful follow-up and for your positive assessment. We are delighted to know that our rebuttal has adequately addressed your concerns, and we sincerely appreciate your support as well as your willingness to raise the score to accept. In the revised version, we will carefully address all issues raised by the reviewers and further strengthen the paper accordingly. We also commit to open-sourcing the full training set, test set, training and evaluation code, as well as the model, to further improve reproducibility and maximize the contribution of this work to the community. Thank you again for your valuable feedback and encouragement.

---

### Decision · Program_Chairs · 2026-04-30

**Decision:**

Accept (regular)

**Comment:**

This paper introduces ViTL (Video-in-the-Loop), a two-stage skim-zoom framework for long-video QA that combines span-grounded learning with joint training of temporal grounding and reasoning, along with the VGrounding-QA dataset. It received one accept and two weak accept ratings. The main concerns are regarding 1) novelty compared to existing zoom-in pipelines; 2) limited benchmark coverage and insufficient experimental details; 3) potential data overfitting and ambiguous architecture gain isolation. After the rebuttal, the reviewers acknowledge that their concerns have been fully addressed through supplementary experiments, detailed explanations, and expanded validation. Reviewer T4zf upgraded to accept, while Reviewers f1sB and 3eoE retained their positive weak accept ratings. Therefore, accept is recommended.